# Olaparib Synergizes the Anticancer Activity of Daunorubicin via Interaction with AKR1C3

**DOI:** 10.3390/cancers12113127

**Published:** 2020-10-26

**Authors:** Tássia S. Tavares, Jakub Hofman, Alžběta Lekešová, Jana Želazková, Vladimír Wsól

**Affiliations:** 1Department of Biochemical Sciences, Faculty of Pharmacy in Hradec Kralove, Charles University, Heyrovskeho 1203, 500 05 Hradec Kralove, Czech Republic; silvatat@faf.cuni.cz (T.S.T.); lekesova@faf.cuni.cz (A.L.); zelazkoj@faf.cuni.cz (J.Ž.); 2Department of Pharmacology and Toxicology, Faculty of Pharmacy in Hradec Kralove, Charles University, Heyrovskeho 1203, 500 05 Hradec Kralove, Czech Republic; hofmanj@faf.cuni.cz

**Keywords:** daunorubicin, olaparib, synergistic, anti-proliferative, leukaemia, KG1α, HepG2

## Abstract

**Sample summary:**

Anthracyclines (ANT) are anti-tumor agents frequently used for the treatment of various cancers. Unfortunately, their clinical success is overshadowed by the emergence of drug resistance. Metabolism by carbonyl reducing enzymes (CREs) represents a critical mechanism of ANT resistance. Here, we have explored possible interactions of CREs with olaparib, an FDA-approved targeted chemotherapeutic. Although olaparib has been demonstrated to potentiate the antiproliferative effect of ANT in experimental models, the causing mechanisms remain unclear. In our study, we demonstrated that olaparib potently inhibits the AKR1C3 reductase at clinically relevant concentrations. Furthermore, we showed that this interaction mediates the reversal of ANT resistance and thus represents a critical mechanism of the synergy between ANT and olaparib. Our observations represent valuable knowledge that could be transformed into the more effective therapy of AKR1C3-expressing tumors.

**Abstract:**

Olaparib is a potent poly (ADP-ribose) polymerase inhibitor currently used in targeted therapy for treating cancer cells with BRCA mutations. Here we investigate the possible interference of olaparib with daunorubicin (Daun) metabolism, mediated by carbonyl-reducing enzymes (CREs), which play a significant role in the resistance of cancer cells to anthracyclines. Incubation experiments with the most active recombinant CREs showed that olaparib is a potent inhibitor of the aldo–keto reductase 1C3 (AKR1C3) enzyme. Subsequent inhibitory assays in the AKR1C3-overexpressing cellular model transfected human colorectal carcinoma HCT116 cells, demonstrating that olaparib significantly inhibits AKR1C3 at the intracellular level. Consequently, molecular docking studies have supported these findings and identified the possible molecular background of the interaction. Drug combination experiments in HCT116, human liver carcinoma HepG2, and leukemic KG1α cell lines showed that this observed interaction can be exploited for the synergistic enhancement of Daun’s antiproliferative effect. Finally, we showed that olaparib had no significant effect on the mRNA expression of AKR1C3 in HepG2 and KG1α cells. In conclusion, our data demonstrate that olaparib interferes with anthracycline metabolism, and suggest that this phenomenon might be utilized for combating anthracycline resistance.

## 1. Introduction

Cancer is one of the most significant health problems worldwide, with 10 million new cases each year [1]. Over the past 60 years, anthracycline family (ANT) drugs have been commonly used to treat various cancers [2]. These chemotherapeutics effectively induce DNA double-strand breaks in rapidly dividing cells. Despite their efficacy, the application of anthracyclines is limited by dose-limiting toxicity to healthy tissues, as well as drug resistance [3]. Numerous efforts have been made to overcome these drawbacks. Nevertheless, diverse mechanisms may mediate ANT resistance, which may include topoisomerase II mutation [4], cancer stemness, DNA repair, overexpression of P-glycoprotein or other efflux pumps, and metabolism [5].

One of the most critical mechanisms of ANT resistance is its reduction to the less potent C13-hydroxy metabolite (e.g., daunorubicinol and doxorubicinol) by carbonyl-reducing enzymes (CREs) [2,6,7]. CREs represent a group of cytosolic or microsomal enzymes comprising two superfamilies: the short-chain dehydrogenase/reductases (SDRs) and aldo–keto reductases (AKRs). Being frequently overexpressed in tumour tissues, CREs decrease the intracellular concentration of pharmacologically active forms of anthracyclines. As a result, this process diminishes their therapeutic efficacy and potentiates cardiotoxic side-effects [3,8]. Among the principal anthracycline reductases (CBR1, AKR1C3, AKR1A1, AKR1B1, and AKR1B10) [2,9], AKR1C3 is the most active enzyme, with a confirmed role in ANT resistance [10]. Its upregulation has been demonstrated in various cancers, indicating its possible value as a diagnostic cancer marker [11,12,13,14]. Importantly, AKR1C3, together with other important CREs (AKR1B10, CBR1), is overexpressed in oncological patients who are resistant to ANTs, which confirms the critical role of reductive metabolism in ANT resistance [2,15,16,17,18]. Considering CREs as essential drivers of ANT resistance and cancer development [14,19,20], as well as the discovery of approaches allowing for modulation of their activity and expression, is of great clinical interest.

Olaparib (AZD2281, trade name Lynparza) is a U.S. Food and Drug Administration (FDA)-approved targeted agent (Figure 1) that acts by preventing DNA damage repair [21,22,23]. This drug is recognized as a potent poly (ADP-ribose) polymerase (PARP) inhibitor working against cancer cells harbouring defects in DNA damage repair (BRCA mutations), which are common in breast, prostate, and ovarian cancers [24]. Beside participation in olaparib’s pharmacodynamic effect, BRCA mutations have a valuable clinical prognostic value, such as guiding the selection of patients eligible for secondary cytoreductive surgery within the treatment of recurrent ovarian cancer with liver metastases [25]. Apart from its common indications, olaparib exhibits significant potential for the treatment of myelodysplastic syndrome and acute myeloid leukaemia (AML) with defects in DNA repair [26,27], inducing the death receptor-mediated apoptosis in AML cells [28,29]. Recently, olaparib was demonstrated to potentiate the effect of doxorubicin in experimental models [30,31]. Additionally, a combination of these two drugs has been evaluated in phase I clinical trials [32]. Despite their promising therapeutic potential, the mechanism underlying this synergism is not yet described in detail. In our work, we hypothesized that olaparib might interference with anthracycline metabolism, which could be one of the molecular mechanisms promoting the beneficial outcome of this combination.

Given these considerations, the present work aimed to characterize the interaction of olaparib with selected recombinant reductases that play a role in anthracycline reduction. Subsequent experiments have evaluated whether these interactions modulate daunorubicin resistance in AKR1C3-expressing models. In the last experimental set, we investigated the possible influence of olaparib on AKR1C3 expression, which can affect the resistance phenotype of cancer cells.

## 2. Results

### 2.1. Screening for Interactions of Anthracycline Reductases with Olaparib

First, we investigated the potential of 10 and 50 µM olaparib to inhibit selected anthracycline reductases extensively involved in daunorubicin metabolism [9,33]. Olaparib potently inhibits AKR1C3, while a negligible inhibitory effect was observed for AKR1B10, AKR1A1, AKR1B1, and CBR1 (Table 1). Considering that olaparib did not display significant inhibition of other anthracyclines reductases, only interactions with AKR1C3 were further investigated at the cellular level.

### 2.2. Olaparib Potently Inhibits Human Recombinant AKR1C3

In subsequent studies with recombinant AKR1C3, we characterized its interaction with olaparib in detail. Olaparib exhibited high inhibitory affinity toward AKR1C3 (half-maximal inhibitory concentration (IC_50_) = 2.48 µM; Figure 2) with a non-competitive inhibition mode (inhibitory constant (*Ki*) = 3.35 µM; α > 1; Figure 3a) as observed in the Lineweaver–Burk analysis plot (Figure 3b).

A docking study was employed to further explore the interaction of olaparib with enzyme structures. The AKR1C3 binding site has already been described in the literature [34], and there are five sub-pockets, referred to as SP1, SP2, SP3, the oxyanion site (OX), and the steroid channel (SC). SP1 is delineated by residues Ser-118, Asn-167, Phe-306, Phe311, and Tyr-319. The majority of non-steroidal anti-inflammatory drugs (NSAIDs), namely N-phenylanthranilic acids (meclofenamic acid, mefenamic acid, and flufenamic acid), arylpropionic acids (flurbiprofen, ibuprofen, naproxen), and zomepirac, bind or extend significantly to this pocket. The NSAID sulindac binds within the SP2 site. Among NSAIDs with known binding modes, only the indomethacin molecule binds the SP3 pocket. However, its binding mode is pH-dependent, and indomethacin can the occupy SP1 site as well. The oxyanion site consists of residues Tyr-55 and His-117, as well as the NADP+ cofactor, and is the catalytic site at which aldehyde or ketone reduction occurs. The carboxyl of NSAIDs form hydrogen bonds to these residues, resulting in the inhibitory effect. The steroid channel represents the open channel that leads to solvent space and is gated by residues Trp-227 and Leu/Val-54 [35].

To predict the binding mode of olaparib, indomethacin, flufenamic acid, and PEG/acetate bound structures (PDB codes 1S2A, 1S2C, and 1S1P, respectively) were used as targets for molecular docking. As already described in the literature, residues Trp-227, Phe-306, and Phe-311 exhibit significant side-chain flexibility depending on the ligand [35]. Thus, docking experiments with side chains of these three residues allowing rotation in the PEG/acetate bound structure (1S1P) were performed as well. The docking of olaparib with flexible residues Trp-227, Phe-306, and Phe-311 provided the best conformations and binding energies. Olaparib occupied the same space as flufenamic acid, with acylhydrazine carbonyl and nitrogen forming hydrogen bonds to the active side (Figure 4a,b).

### 2.3. Effect of Olaparib on AKR1C3-Mediated Daunorubicin Metabolism in HCT116 Cells

To evaluate the inhibitory effect of olaparib on daunorubicin (Daun) reduction at the cellular level, an inhibitory assay in AKR1C3 and empty vector-transfected HCT116 cells (HCT116–AKR1C3 and HCT116–EV (empty vector), respectively) was employed. A clinically relevant Daun concentration (1 µM) was used in these experiments [36].

In HCT116–AKR1C3 cells, olaparib showed significant dose-dependent inhibition of AKR1C3-mediated Daun metabolism (IC_50_ = 5.91 µM; Figure 5). These data correlate well with those from the recombinant AKR1C3 enzyme and demonstrate the ability of the tested drug to interfere with AKR1C3-mediated daunorubicin metabolism at the cellular level.

### 2.4. Olaparib Synergize with Daunorubicin due to Interaction with AKR1C3

Having confirmed AKR1C3 inhibition by olaparib at intracellular conditions, we next aimed to investigate whether this interaction could reverse Daun resistance. At the same time, we examined whether it might be one of the hidden mechanisms underlying the reported efficacy of combined anthracycline + olaparib.

In our combination studies in transfected HCT116 cells, we used clinically relevant concentrations of Daun [36] and olaparib [37]. Besides being clinically relevant, olaparib concentrations also (1) yielded sufficient inhibition of AKR1C3 and (2) exerted negligible toxicity in tested cell lines (IC_50_ was equal to 202 ± 46 µM and 186 ± 51 µM in HCT116–AKR1C3 and HCT116–EV cells, respectively). In accordance with the recognized role of AKR1C3 in Daun resistance, the AKR1C3-overexpressing subline showed reduced sensitivity to Daun compared to the EV-transduced subline (IC_50_ values of 0.808 ± 0.049 µM and 0.494 ± 0.039 µM, respectively). Olaparib potentiated the effect of Daun in both HCT116 sublines, with a trend toward increased effects in the AKR1C3-overexpressing cell line (Figure 6a,b). Since the results of the combination are affected by olaparib toxicity, we performed data analysis using the Chou–Talalay method, which subtracts this distorting element and accurately quantifies combination outcomes. At 5 µM olaparib concentration, the combination effect in EV cells was additive or antagonistic, while in AKR1C3-overexpressing cells, synergism was detected along the entire range of cells affected (F_A_) (Figure 6c). At 10 µM, synergism was observed in both cell lines, but differences in the effect between cell sublines was evident (Figure 6d). To exclude a possible interfering effect of differential PARP expression on combination outcomes, we performed assessment of *PARP1* and *PARP2* (pharmacodynamic targets of olaparib [38,39]) expression in transfected HCT116 cells. As there were no expression changes (Figure 6e), we can conclude that the contribution of PARP inhibition to the combination outcome was identical in both cell sublines. In turn, this observation confirms the hypothesis that differences in combination efficacy between sublines are due to interactions with AKR1C3.

To clarify the possible clinical impact of these results, additional experiments were conducted with cells possessing physiological expressions of AKR1C3 [40]. Combining 5 or 10 µM olaparib with Daun resulted in a synergistic outcome along the major part of F_A_ in KG1α leukemic cells (Figure 7a,c). In HepG2 cells, the synergistic effect was primarily evident at higher F_A_ fractions in response to 1 uM Daun (Figure 7b,d). Taken together, these results demonstrate that olaparib-mediated inhibition of AKR1C3 can, at least partially, reverse Daun resistance.

### 2.5. Assessment of AKR1C3 Expression following Exposure to Olaparib

In the final study, we performed quantitative reverse transcription real-time PCR (qRT-PCR) analysis to assess possible changes in *AKR1C3* mRNA expression following exposure to olaparib. First, experiments conducted in HepG2 liver carcinoma cells, representing a systemic model, were employed to determine whether olaparib has the potential to affect whole-body pharmacokinetics of Daun. Second, KG1α leukemic cells were used as a tumoral model to determine whether the positive synergistic olaparib + anthracycline effect could be counteracted by enzyme induction.

First, we evaluated the effect of olaparib on model cell viability to choose a drug concentration with tolerable cytotoxicity. The selected clinically relevant olaparib concentrations of 1 and 5 µM displayed low cytotoxicity levels in both cell lines (Figure 8a,b). In the induction experiments, olaparib did not provoke any significant changes in *AKR1C3* expression in either HepG2 or KG1α cells (Figure 8c,d). The absence of AKR1C3 induction leads to the assumption that olaparib does not influence Daun’s metabolism, and does not even threaten the combination effect via the upregulation of AKR1C3 expression.

## 3. Discussion

Olaparib, marketed under the trade name Lynparza, is a potent, FDA-approved PARP inhibitor (PARPi) for the treatment of ovarian, breast [41], and pancreatic [42] cancers. Olaparib’s mechanism of action follows a concept of synthetic lethality: this drug selectively targets cancer cells with hereditary BRCA1/2 mutations [24]. It has been recently shown that olaparib potentiates the anticancer activity of anthracyclines [30,31]. However, the molecular mechanism of this beneficial effect is poorly understood. In this study, we investigated possible interactions of olaparib with anthracycline reductases, and evaluated whether they could reverse daunorubicin resistance and participate in the positive outcomes observed in this combination.

Olaparib’s interaction with AKR1C3 was observed at the recombinant enzyme and cellular level, reaching roughly similar strengths of low micromolar concentrations in both variants. Importantly, the average maximal plasma concentration (*C_max_*) of olaparib in patients reached 14.2 µM at the recommended dosage of 400 mg [37]. Considering the relatively low extent of olaparib’s plasma protein binding (≈ 82%) [37], as well as EMA (European Medicines Agency) guidelines for in vitro testing of drug–drug interactions on biotransformation enzymes, it is highly probable that this recorded interaction could manifest in oncology patients. According to the instructions, interactions with drugs exhibiting *Ki* ≤ 50-fold unbound fractions of *C_max_* are clinically relevant [43].

Having confirmed olaparib-mediated inhibition of AKR1C3 at the cellular level, we presumed that this process could help reverse resistance to anthracycline. Our results demonstrated a differential response in AKR1C3- vs. empty vector-transfected HCT116 cells to the Daun combination with olaparib. Thus, this hypothesis was confirmed, suggesting that this mechanism is likely to participate in the overall synergistic outcome of this combination [32]. Synergism was also observed in KG1α and HepG2 cells, which physiologically express considerable amounts of AKR1C3. This finding implies that the conclusion of the artificial HCT116–AKR1C3 model might be applicable to the clinical situation as well.

In addition to its application in combination with anthracyclines, olaparib’s interaction with AKR1C3 might play an important role on its own. AKR1C3 catalyses the synthesis of pro-proliferative prostaglandins that suppresses myeloid and erythroid differentiation [11,12]. In addition, AKR1C3 participates in biochemical pathways leading to the synthesis of dihydrotestosterone (DHT), a potent endogenous ligand of the androgen (AR) receptor [44,45]. In the classical pathway, AKR1C3 catalyses the conversion of 4-androstenedione (Δ4-AD) into testosterone (T) (dehydroepiandrosterone (DHEA) → Δ4-AD → T → DHT), also converting 5α-Adione into DHT (DHEA → Δ4-AD → 5α-Adione → DHT), and androsterone into 3α-Diol (progesterone → 5α-dihydro-progesterone → allopregnanolone → androsterone → 3α-Diol → DHT) in alternative pathways [44]. In general, AKR1C3 expression and AR-signalling are related to the development of several malignancies (e.g., bladder cancer, renal cell carcinoma, hepatocellular cancer, and endometrial cancer), as well as ovarian, breast, and pancreatic cancers [45,46], all of which are therapeutic targets of olaparib. Considering these issues, our findings suggest that olaparib might antagonize AR signaling and the synthesis of pro-proliferative prostaglandins. Thus, the interaction of olaparib with AKR1C3 could be a side-mechanism of action in both hormone-dependent and -independent cancers. This hypothesis warrants further investigation.

## 4. Materials and Methods

### 4.1. Reagents and Chemicals

Olaparib was obtained from SelleckChem (Houston, TX, USA). Glucose-6-phosphate dehydrogenase and JetPrime Polyplus transfection reagent were acquired from VWR International Ltd., whereas daunorubicin (Daun) was purchased from Toronto Research Chemicals (Toronto, ON, Canada). TRI Reagent solution was acquired from Molecular Research Center Inc. (Cincinnati, OH, USA). Nicotinamide adenine dinucleotide phosphate (NADP^+^), glucose-6-phosphate, XTT (2,3-Bis-(2-Methoxy-4-Nitro-5-Sulfophenyl)-2H-Tetrazolium-5-Carboxanilide), phenazine methosulfate (PMS), foetal bovine serum (FBS), and HPLC-grade solvents were supplied by Sigma-Aldrich (Prague, Czech Republic). Cell culture reagents were acquired from Lonza (Walkersville, MD, USA) and Sigma Aldrich (St. Louis, MO, USA). The RNA extraction kit was purchased from Zymo Research (Irvine, CA, USA). *AKR1C3*-specific primers, together with the oligo-dT primer and *PARP1* and *PARP2* qRT-PCR standards, were acquired from Generi Biotech (Hradec Králové, Czech Republic). *PARP1* and *PARP2* TaqMan systems and TaqMan Universal Master Mix II (no UNG) were obtained from Thermo Fisher Scientific (Indianapolis, IN, USA). The qPCR SG Mix was from the Institute of Applied Biotechnologies (Prague, Czech Republic). ProtoScript II Buffer, DTT, dNTP mix, and ProtoScript II Reverse Transcriptase were obtained from New England Biolabs (Ipswich, MA, USA). All reagents used were of the highest commercially available purity.

### 4.2. Cell Cultures

Human colorectal carcinoma cells (HCT116), together with human liver carcinoma (HepG2) and leukemic (KG1α) cell lines, were acquired from the European Collection of Authenticated Cell Cultures (ECACC, Salisbury, UK). HCT116 and HepG2 cells were grown in DMEM (Dulbecco’s modified Eagle’s medium) supplemented with 10% foetal bovine serum, whereas KG1α was grown in IMDM (Iscove’s Modified Dulbecco’s Medium), supplemented with 20% foetal bovine serum and 2 mM L-glutamine at standard conditions (37 °C, 5% CO_2_). All experiments and routine cultivation were conducted in an antibiotic-free medium. Cell lines were used at passages between 10–25 and were periodically checked for mycoplasma infection. The olaparib solution was prepared in dimethyl sulfoxide (DMSO), such that solvent concentrations did not exceed 0.5% *v/v*. The vehicle control approach was used to eliminate possible distorting effects of the drug solvent on examined parameters.

### 4.3. Cloning, Overexpression, and Purification of Recombinant CREs

According to previous publications [10,47,48], human recombinant CREs (AKR1B10, AKR1C3, AKR1A1, AKR1B1, and CBR1) were prepared in the *E. coli* BL21 (DE3) host system. CRE purification was performed by affinity chromatography using the NGC chromatography low-pressure system, which was equipped with a 1 mL HisTrap FF column (GE Healthcare Life Sciences, Marlborough, MA, USA) acquired from Sigma-Aldrich (Prague, Czech Republic). Buffer A was composed of 20 mM Tris-HCl, 150 mM NaCl, 20% (*v/v*) glycerol, and 30 mM imidazole (pH 7.4). Buffer B was identical to buffer A, except that the imidazole concentration was 500 mM. The supernatant was adjusted to contain 30 mM imidazole and 500 mM NaCl before being loaded onto the column. The purification was performed as follows: (1) the sample was loaded onto the column, (2) washed with 5 mL of buffer A, and (3) pure protein was eluted by increasing concentrations of buffer B (60% in 20 min). Finally, the column was regenerated by 10 mL of buffer B, and the active fractions containing the purified protein were pooled. Specific enzymatic activity (µmol/mg/min) was calculated based on the rate of Daun reduction for the formation of daunorubicinol (Daun-ol), as described below. The enzymatic solution (20 mM phosphate buffer, pH 7.4) was supplemented with glycerol (final concentration 20%), and the protein concentrated to 1.5 mg/mL.

### 4.4. Incubation Assay with Recombinant CREs

All enzymatic activity assays (AKR1C3, AKR1B10, AKR1A1, AKR1B1, and CBR1) were performed in 0.1 M phosphate buffer containing 1.5 µg of protein per reaction, Daun substrate (500 µM), and NADP^+^ regeneration system (2.6 mM NADP^+^, 19.2 mM glucose-6-phosphate, 0.34 U glucose-6-phosphate dehydrogenase, 9.8 mM MgCl_2_, and 0.1 M phosphate buffer at pH 7.4) were stirred continuously for 30 min at 37 °C. CRE inhibitory assays were performed with olaparib (10 and 50 µM). The reaction was stopped with NH_4_OH (26%), and Daun-ol was then isolated by two-step extraction into ethyl acetate using shaking for 15 min, following centrifugation for 2 min (13,500 rpm). The organic phase was evaporated under vacuum, and residuum was dissolved in mobile phase and subjected to UHPLC (ultra-high-performance liquid chromatography). The inhibitory action of olaparib and its affinity for AKR1C3 was assessed by experimental IC_50_ (half-maximal inhibitory concentration) determination, using olaparib concentrations a range of 0.01–50.00 μM. Inhibition assays were conducted using Daun concentrations of 500, 750, 1000, and 1250 μM. Employing GraphPad Prism 8.0.4, Michaelis–Menten parameters were calculated and transformed in a Lineweaver–Burk double reciprocal plot to determine the mode of inhibition by analysing the slopes and intercepts of the curves.

### 4.5. Docking Studies

AutoDock Vina [49] software was used to perform all docking experiments. Each docking experiment was run at least twice. The X-ray structures of AKR1C3 (indomethacin cocrystal (1S2A [3])), AKR1C3 (flufenamic acid cocrystal (1S2C [3])), and AKR1C3 (containing PEG and acetate in the active site, 1S1P [3]) were used in this study. Their structures were prepared for docking using AutoDockTools [50]. All water was removed, all small molecules except for NADP were deleted, polar hydrogens were added, Kollman charges were calculated, and structures were exported into pdbqt format. The docking cube position and its dimensions were determined visually using AutoDockTools. The dimensions of docking cube were set to 30 Å × 30 Å × 30 Å in all cases. The docking space coordinates *x* = 28.911, *y* = −26.519, and *z* = 59.43 were set for all targets (1S1P and 1S2A). The exhaustiveness parameter was set based on 64 in experiments with the rigid receptor, while it was increased to 128 for experiments with flexible receptor residues. In these experiments, residues Trp-227, Phe-306, and Phe-311 were allowed to rotate around Cα–Cβ and Cβ–Cγ. The ligand olaparib was downloaded as mol2 files from the ZINC database [51] and converted to pdbqt files using AutoDockTools. For model evaluation, ligands extracted from original X-ray structures were exported to pdbqt format and docked. Docking study results were inspected visually using Discovery Studio Visualizer [52], which was also used to create figures representing the binding interactions.

### 4.6. Transient Transfection of HCT116 Cell Line

The pCI_AKR1C3 plasmid encoding the AKR1C3 enzyme and pCI Empty Vector (EV) was propagated in the *E. coli* system as previously described Chou–Talalay method]. HCT116 cells (3.0 × 10^5^ cells/well) were seeded into a 24-well plate for 24 h. After reaching a confluence of 60%, the transfection mixture containing 0.25 µg of DNA (diluted in Opti-MEM medium) and 0.75 μL of JetPrime transfection reagent was incubated for 10 min at room temperature, according to the supplier’s instructions. Meanwhile, culture media was replaced with a fresh supply. The transfection solution was added dropwise into the wells, and transfected cells were incubated for 24 h before being used for AKR1C3 inhibitory studies or drug combinations. The uniformity of transfection and AKR1C3 expression were monitored as previously described [53].

### 4.7. AKR1C3 Inhibitory Assay in HCT116 Cells

The AKR1C3 inhibitory activity of olaparib was assessed at the cellular level. Transfected HCT116 cells were established according to the procedure described above. Media containing 5 μM Daun, with or without olaparib solution, was added to each well (final concentrations: 1, 5, 10, 25, and 50 μM). The drug solutions and vehicle controls were harvested after 2 and 4 h incubation at standard conditions (37 °C, 5% CO_2_). Thereafter, cells were lysed with 200 μL lysing buffer (25 mM Tris, 150 mM NaCl, 1% Triton X-100, pH 7.8) for 15 min at room temperature. Drug solutions and vehicle controls were mixed with respective lysates, and formed Daun-ol was extracted twice using ethyl acetate. Following evaporation of the organic phase under the vacuum and dissolution of the dry residues in the mobile phase, Daun-ol concentrations were determined by UHPLC.

### 4.8. XTT Proliferation Assay

This method was used to assess cell viability in drug combinations and prior induction studies (see next subsections). Cells were treated with drugs or a vehicle for the time intervals specified in their respective subsections. Cellular proliferation was assessed using a detection solution composed of XTT (1 mg/mL) in Opti-MEM medium and phenazine methosulphate (7.5 mg/mL). The HCT116, HepG2, and KG1α cells were incubated with the detection solution for 1 h, 2 h, and 3 h, respectively. Cellular absorbance was measured at 450 nm using a microplate reader (Infinite M200, Tecan, Salzburg, Austria).

### 4.9. Drug Combination

Transfected HCT116 (2.5 × 10^4^ cells/well) and HepG2 (1.8 × 10^4^ cells/well) cells were seeded on 96-well plates. After 24 h, transfected HCT116 cells and HepG2 cells were incubated with Daun (range of 0.01–1.00 µM) with or without 5 or 10 µM olaparib. As well as the combination, the toxicity of olaparib alone was also assessed (ranges of 0.5–200.0 µM and 0.1–100.0 µM for transfected HCT116 and HepG2, respectively). KG1α (2.5 × 10^4^ cells/well) cells were seeded and immediately treated with Daun (0.010–0.250 µM) with or without 5 or 10 µM olaparib. In addition, olaparib toxicity alone was assessed in a range of 0.05–50.00 µM. For all cell lines, viability was determined after 72 h of incubation at standard conditions using an XTT assay (see the previous subsection). Combination effects were quantified according to a combination index (CoI) from the Chou–Talalay method [53]. Next to drug combinations, *PARP1* and *PARP2* mRNA expression was monitored in transfected HCT116 sublines using qRT-PCR. Absolute expression of target genes was quantified using TaqMan probes, TaqMan Universal Master Mix II (no UNG), and qRT-PCR standards, according to manufacturer’s instructions. RNA isolation and cDNA preparation were performed as described in the following section.

### 4.10. Induction Studies

First, olaparib cytotoxicity was screened in model cell lines to estimate a suitable concentration for use in induction studies. HepG2 and KG1α cells were seeded at densities identical to those used in the drug combination studies. Cells were treated with olaparib on a six-point concentration range (0.05–20.00 µM) and incubated for 48 h at standard conditions. Cell viability was assessed using an XTT proliferation assay. Subsequently, induction studies based on qRT-PCR analysis were performed to determine the effect of olaparib on *AKR1C3* expression. HepG2 (30 × 10^4^) and KG1α (25 × 10^4^) were seeded into 12-well plates (KG1α at time 0, HepG2 24 h before time 0). At time 0, cells were treated with 1 or 5 µM olaparib or vehicle control; then, samples were collected at 24 and 48 h intervals. For RNA extraction, cells were lysed with TriReagent, and total RNA was extracted using Zymo Research’s Direct-Zol RNA mini prep kit. The cDNA was generated as described previously [54]. *AKR1C3* mRNA in samples was quantified using *AKR1C3* standard, which was prepared as described previously [10]. The qRT-PCR was performed by setting up the following reactions: 1x concentrated XCEED qPCR SG Mix, *AKR1C3*-specific primers (1 µM), and 20 ng cDNA. The qPCR was performed using a QuantStudio 6flex (Applied Biosystems by Life Technologies, Carlsbad, CA, USA) with the following conditions: initial denaturation at 95 °C for 10 min, then 40 cycles at 95 °C for 15 s and 65 °C for 1 min.

### 4.11. Ultra-High-Performance Liquid Chromatography (UHPLC)

Daun-ol concentrations were determined using a UHPLC Agilent 1260 Series chromatographic system equipped with a dual-fluorescence detector (Shimadzu, Japan). The enzyme reaction mixture was pre-filtered through a 0.2 μm polytetrafluoroethylene (PTFE) syringe filter, and then injected into a Zorbax Eclipse Plus C18 RR HD column (2.1 × 50 mm, 1.8 μm particle size) with a 1290 Infinity inline filter (Agilent, Santa Clara, CA, USA). The mobile phase consisted of a mixture containing formic acid (0.1%), water (74%), and acetonitrile (26%). The flow rate was 0.7 mL/min. Eluents were detected using a fluorescence detector at excitation and emission wavelengths of 480 and 560 nm, respectively.

### 4.12. Statistical Analysis

Statistical analysis was performed in GraphPad Prism 8.0.4 (GraphPad Software, Inc., La Jolla, CA, USA); *p*-values < 0.05 were considered statistically significant. One-way ANOVA followed by Dunnett’s post-hoc test or a two-tailed unpaired *t*-test were used for *p*-value calculations, as specified in particular figure legends. Combination indices (CoI) were calculated using CompuSyn 3.0.1 software (ComboSyn Inc., Paramus, NJ, USA).

## 5. Conclusions

In conclusion, we demonstrated that olaparib potently inhibits the AKR1C3 enzyme at clinically relevant concentrations. This interaction was proven to attenuate Daun resistance, and can be assumed to participate on the synergistic outcome of olaparib combined with anthracyclines. Advantageously, olaparib did not affect *AKR1C3* expression in either systemic or intratumoral models, and thus does not exhibit the potential to establish systemic Daun resistance or counteract the beneficial combination effect. Our observations represent valuable knowledge that could be transformed into more effective therapies in AKR1C3-expressing tumours.

## Figures and Tables

**Figure 1 cancers-12-03127-f001:**
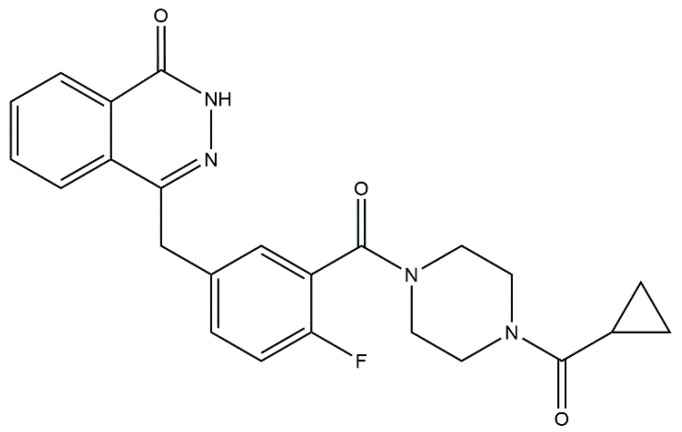
Chemical structure of olaparib.

**Figure 2 cancers-12-03127-f002:**
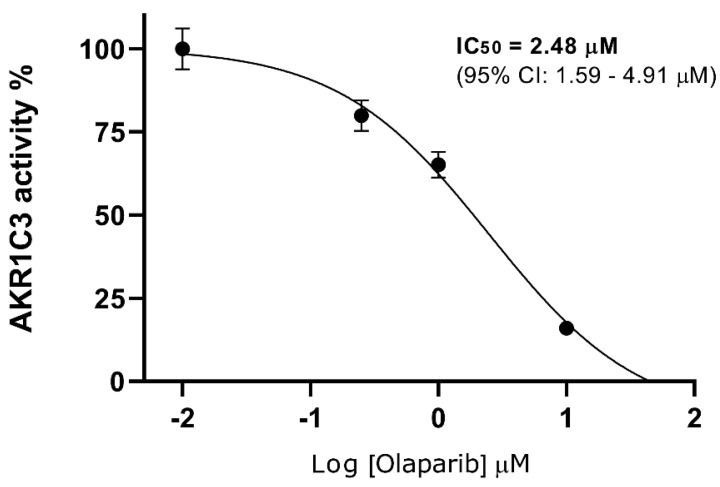
IC_50_ (half-maximal inhibitory concentration) of AKR1C3 inhibition by olaparib. Data are expressed as the mean ± SD from three independent assays.

**Figure 3 cancers-12-03127-f003:**
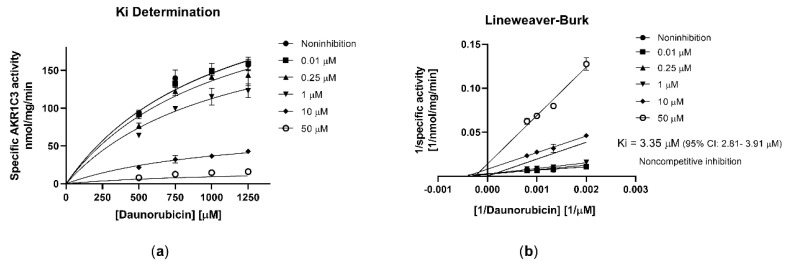
(**a**) Inhibitory constant (*Ki)*-value determination. (**b**) Lineweaver–Burk double-reciprocal plot. Data are expressed as the mean ± SD from three independent assays. Measurements were performed using different substrate concentrations.

**Figure 4 cancers-12-03127-f004:**
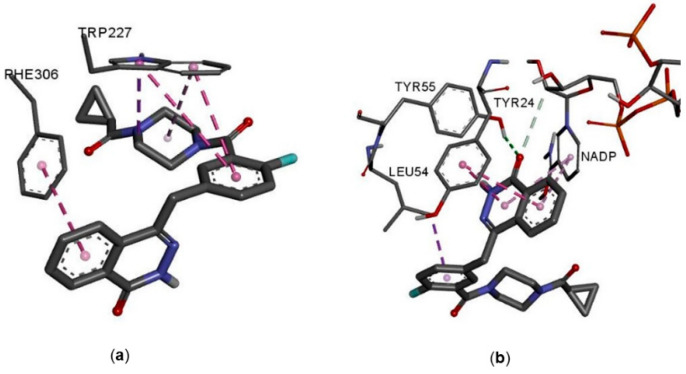
Interactions of olaparib with the AKR1C3 flexible residues. (**a**) Rigid residues and NADP. (**b**) Amino acid residues involved in interactions with a ligand are rendered as sticks. Ligands are rendered as double-sized sticks. Carbon is illustrated in grey, nitrogen in blue, oxygen in red, fluorine in cyan, phosphorus in orange, and hydrogen in white. Only polar hydrogens are shown. H-bonds are depicted as green dashed lines, and hydrophobic interactions as violet dashed line (π–σ in dark violet, π–π in pink, and π–alkyl in light pink).

**Figure 5 cancers-12-03127-f005:**
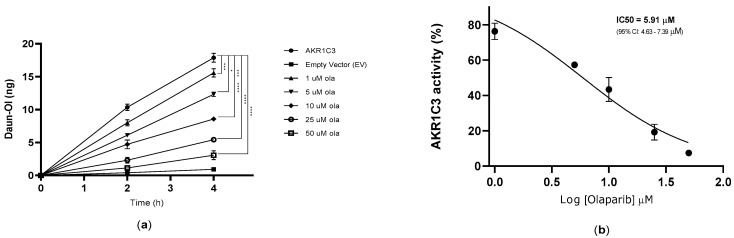
(**a**) Effect of olaparib on AKR1C3-mediated intracellular daunorubicin (Daun) metabolism in HCT116 cells. Statistical analysis was performed with end-point data, using one-way analysis of variance (ANOVA) followed by Dunnett’s post-hoc test (* *p* ≤ 0.05, *** *p* ≤ and **** *p* ≤ 0.0001, compared to AKR1C3 control). (**b**) IC_50_ of AKR1C3 inhibition in HCT116 cells exposed to olaparib (1–50 µM). Data are expressed as the mean ± SD from three independent assays.

**Figure 6 cancers-12-03127-f006:**
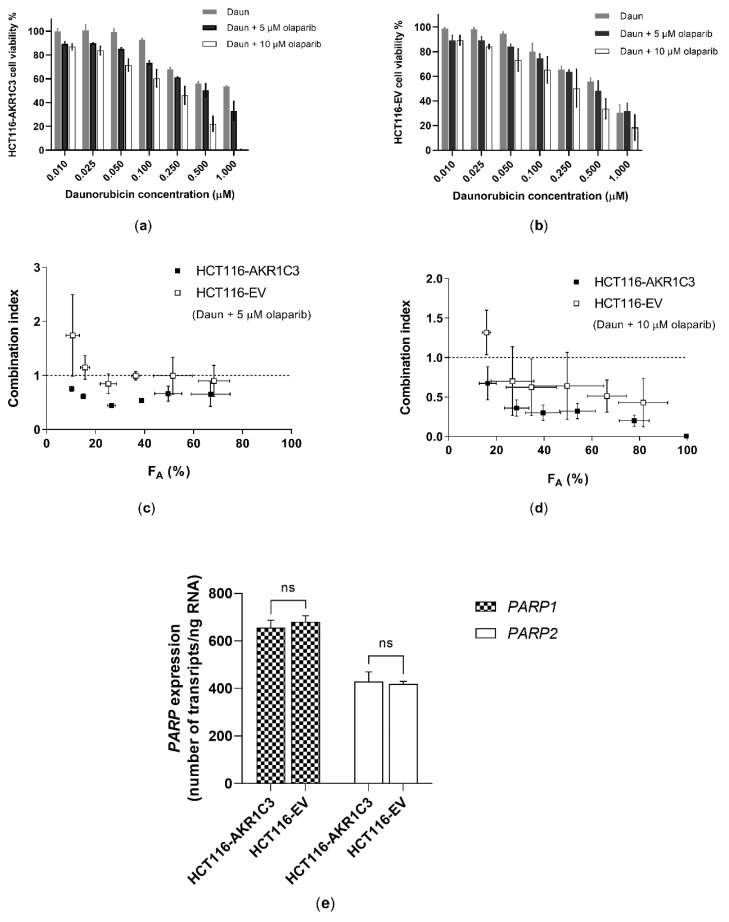
Effect of olaparib combined with Daun in transfected (**a**) HCT116–AKR1C3 and (**b**) HCT116–EV (empty vector) subline. GraphPad Prism 8.4.0 was used to perform the normalization of absorbance values; 0% and 100% were assigned to cells incubated with 10% DMSO and vehicle control, respectively. Data are expressed as the mean ± SD from three independent assays. Daun alone, olaparib alone, and their combinations were further analysed by the Chou–Talalay method, generating combination indices (CoI). CoI < 0.9 represents synergism, CoI > 0.9 and <1.1 represents additivity, and CoI > 1.1 represents antagonism. Cells affected (F_A_)–CoI plot of (**c**) 5 µM olaparib with Daun and (**d**) 10 µM olaparib with Daun. (**e**) *PARP1* and *PARP2* expression was quantified using quantitative reverse transcription real-time PCR (qRT-PCR) in transfected HCT116 cells. The expression data were statistically analysed using a two-tailed unpaired *t*-test. Data are presented as the mean ± SD from three independent transfections.

**Figure 7 cancers-12-03127-f007:**
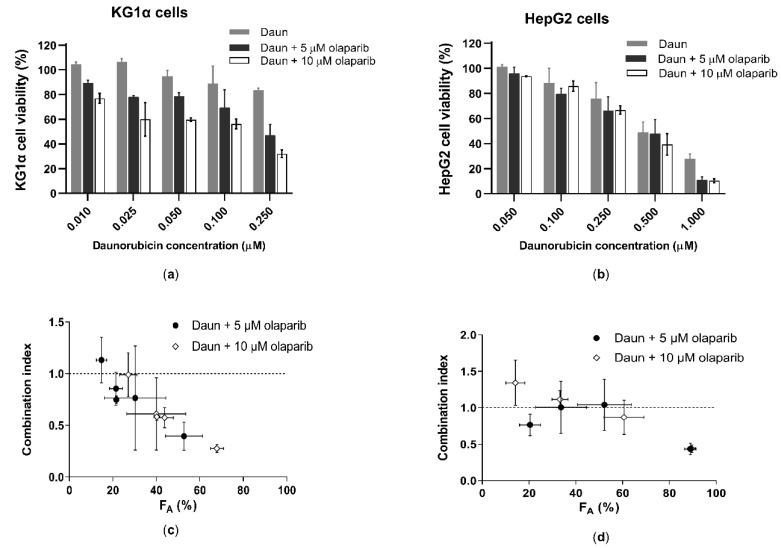
Effect of olaparib combination with Daun in (**a**) KG1α cells and (**b**) HepG2 cells. GraphPad Prism 8.4.0 was used to perform normalization of absorbance values; 0% and 100% were assigned to cells incubated with 10% DMSO and vehicle control, respectively. Data are expressed as the mean ± SD from three independent assays. Daun alone, olaparib alone, and their combinations were further analysed by the Chou–Talalay method, generating combination indices (CoI). CIS < 0.9 represents synergism, CoI > 0.9 and <1.1 represent additivity, and CoI > 1.1 represents antagonism. F_A_–CoI plot of (**c**) 5 µM or 10 µM of olaparib with Daun in KG1α cells, as well as (**d**) 5 µM or 10 µM of olaparib with Daun in HepG2 cells.

**Figure 8 cancers-12-03127-f008:**
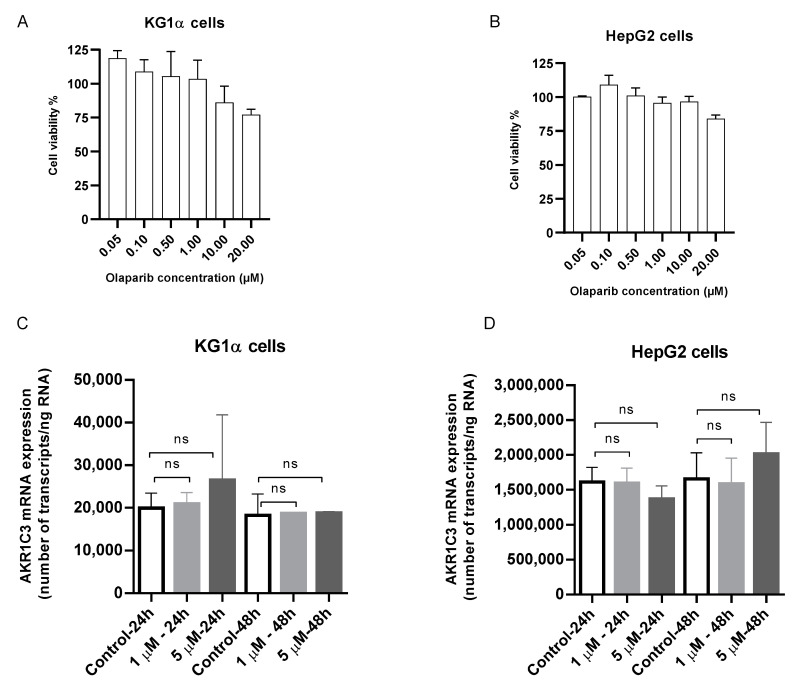
Effect of olaparib (0.05–20.00 µM) on cell viability in (**a**) KG1α cells and (**b**) HepG2 cells after 48 h exposure. Cell viability of 0% and 100% were assigned to cells incubated with 10% DMSO and vehicle control, respectively. GraphPad Prism 8.4.0 was used to perform normalization of absorbance values. Determination of *AKR1C3* expression after exposure to olaparib (1 and 5 µM) in (**c**) KG1α cells and (**d**) HepG2 cells. After 24 h and 48 h, possible changes in *AKR1C3* mRNA expression were monitored by quantitative reverse transcription real-time PCR (qRT-PCR). Student’s *t*-test (unpaired) was employed to assess statistical significance (ns: not significant). Data are presented as the means ± SD of three independent experiments.

**Table 1 cancers-12-03127-t001:** Inhibitory effect of olaparib on anthracycline reductases.

Enzymes	Inhibition % Olaparib (10 µM)	Inhibition % Olaparib (50 µM)
AKR1C3	76.0 ± 0.32	91.4 ± 0.97
AKR1B10	0.0	7.7 ± 0.33
AKR1A1	1.8 ± 0.14	18.2 ± 0.39
AKR1B1	0.0	8.1 ± 0.83
CBR1	0.0	0.0

Aldo–keto reductase (AKR) and Carbonyl reductase (CBR). Values are expressed as the means ± SD of three independent assays.

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
