# Peer review of "Olaparib Synergizes the Anticancer Activity of Daunorubicin via Interaction with AKR1C3"

_cancers, 2020, doi:10.3390/cancers12113127_

Round 1

Reviewer 1 Report

The authors fulfilled my requirements from the first review. I think that now is manuscript in the form which is publishable.

Author Response

nothing to reply

Reviewer 2 Report

Good work

Author Response

nothing to reply

Reviewer 3 Report

Authors conducted extra experiments for my suggestion. It was not exactly what I suggested but their new experiments supported their story and are acceptable. I am satisfied with their comments.

Author Response

nothing to reply

This manuscript is a resubmission of an earlier submission. The following is a list of the peer review reports and author responses from that submission.

Round 1

Reviewer 1 Report

The manuscript submitted by Tavares et al showed olaparib's another target for anticancer effect. They showed olaparib inhibit AKR1C enzyme and molecular pathways for anticancer mechanisms.

I really enjoyed reading this paper and was surprised to see another target of olaparib. Authors used inhibition of AKR1C3 in overexpression model to observe anticancer effect. Authors demonstrated overexpression cells showed anticancer effect with Daun. But I really want to see the effect in PARP knockout cells. It is still difficult to eliminate the effect of PARP by olaparib. By this experiments, authors can claim their theory without problems. 

Author Response

We thank the Reviewer for the interest in our work and for helpful comments that will significantly improve the manuscript, and we have tried to do our best to respond to the points raised. The Reviewer has brought up some good points, and we appreciate the opportunity to clarify our research objectives and results. As indicated below, we have checked the general and specific comments provided by the Referee and have made necessary changes accordingly to the indications.

General comment:

C: The manuscript submitted by Tavares et al showed olaparib's another target for anticancer effect. They showed olaparib inhibit AKR1C enzyme and molecular pathways for anticancer mechanisms.

I really enjoyed reading this paper and was surprised to see another target of olaparib. Authors used inhibition of AKR1C3 in overexpression model to observe anticancer effect. Authors demonstrated overexpression cells showed anticancer effect with Daun. But I really want to see the effect in PARP knockout cells. It is still difficult to eliminate the effect of PARP by olaparib. By this experiment, authors can claim their theory without problems.

A: Thank you for your commentary. We agree that PARP inhibition, as a main mechanism of olaparib´s action, for sure affects the outcome of olaparib + anthracycline combination. On the other hand, in our study, we are not able to quantify the participation of all possible mechanisms of synergism between these drugs as there can be many (unknown) ones. The main aim of our study (which is presented in more places in the manuscript) was to clarify, whether olaparib-mediated interference with anthracycline metabolism could be one of the mechanisms hidden beyond the beneficial outcome of olaparib + anthracycline combination observed in recent experimental studies (Park et al., 2018; Eetezadi et al., 2018)*. Comparison of combination outcome in HCT116-EV and HCT116-AKR1C3 is crucial to answer this hypothesis. These cells should differ between each other only in the expression of AKR1C3 and are, thus, ideal model for this purpose. In our combination studies, we observed higher effectivity of drug combination in HCT116-AKR1C3 than in HCT116-EV cells (Fig. 6), which clearly demonstrates the participation of AKR1C3-inhibitory effect in affecting (improving) combination outcome. Unfortunately, we are not technically and mentally equipped to prepare gene-knockout cellular models. However, to comply with your suggestion and to eliminate the doubts about possible distorting effect given by differential level of PARP inhibition, we performed alternative additional experiments. In these experiments, we determined, whether PARP1 and PARP2 expression might be changed following transfection of HCT116 cells. Results of these qRT-PCR studies showed comparable expression of PARP1 and PARP2 in both cell sublines. It is, therefore, possible to presume that the contribution of PARP inhibition into combination outcome is identical in both sublines and observed differences in combination efficacy between HCT116-EV and HCT116-AKR1C3 are indeed brought by the interaction on AKR1C3. We believe that this alternative approach properly adress your concern regarding contribution of PARP inhibition to the combination outcome. The method together with results (new subpicture Fig. 6e) and respective text were introduced into revised manuscript (changes are made in Tracking of changes mode).

Figure 6. “…(e) PARP1 and PARP2 expression was quantified using quantitative reverse transcription real-time PCR (qRT-PCR) in transfected HCT116 cells. Data are presented as the mean ± SD from three independent transfections.”

*References:

  1. Park, H.J.; Bae, J.S.; Kim, K.M.; Moon, Y.J.; Park, S.H.; Ha, S.H.; Hussein, U.K.; Zhang, Z.; Park, H.S.; Park, B.H.; et al. The PARP inhibitor olaparib potentiates the effect of the DNA damaging agent doxorubicin in osteosarcoma. J. Exp. Clin. Cancer Res. 2018, 37, 107, doi:10.1186/s13046-018-0772-9.
  2. Eetezadi, S.; Evans, J.C.; Shen, Y.T.; De Souza, R.; Piquette-Miller, M.; Allen, C. Ratio-Dependent Synergism of a Doxorubicin and Olaparib Combination in 2D and Spheroid Models of Ovarian Cancer. Mol. Pharm. 2018, 15, 472–485, doi:10.1021/acs.molpharmaceut.7b00843.

Reviewer 2 Report

the combination of anticancer drugs with other  targeted drugs represents the most important challenge of translational research. The present study investigates the potential role of olaparib to reduce the chemoresistance, and this combination could be introduce in the clinical practice for the treatment of ovarian cancer. It can therefore constitute a field of research and clinical application in the future. 

Obviously, translational research can be very complex because in clinical practice multiple factors can influence the effectiveness of the therapies adopted. In this sense, personalized medicine plays a fundamental role.  For example, the knowledge of some clinical behaviors, on the basis of the molecular characteristics of the tumor, influence the clinical behavior. It may be useful to refer, for example, to these articles in which we see how in clinical practice the molecular and genetic characteristics influence the prognosis of patients with ovarian cancer:

Prognostic factors value of germline and somatic brca in patients undergoing surgery for recurrent ovarian cancer with liver metastases Eur J Surg Oncol. 2019 Nov;45(11):2096-2102

Author Response

We thank the Reviewer for the interest in our work and for helpful comments that will significantly improve the manuscript, and we have tried to do our best to respond to the point raised. The Reviewer has brought up some a good point, and we appreciate the opportunity to clarify our research. As indicated below, we have checked the comment provided by the Referee and have made necessary changes accordingly to the indications.

General comment:

C: The combination of anticancer drugs with other targeted drugs represents the most important challenge of translational research. The present study investigates the potential role of olaparib to reduce the chemoresistance, and this combination could be introduce in the clinical practice for the treatment of ovarian cancer. It can therefore constitute a field of research and clinical application in the future.

Obviously, translational research can be very complex because in clinical practice multiple factors can influence the effectiveness of the therapies adopted. In this sense, personalized medicine plays a fundamental role.  For example, the knowledge of some clinical behaviors, on the basis of the molecular characteristics of the tumor, influence the clinical behavior. It may be useful to refer, for example, to these articles in which we see how in clinical practice the molecular and genetic characteristics influence the prognosis of patients with ovarian cancer:

Prognostic factors value of germline and somatic brca in patients undergoing surgery for recurrent ovarian cancer with liver metastases Eur J Surg Oncol. 2019 Nov;45(11):2096-2102

A: Thank you for your comment. We have added the suggested information and respective reference.

P.2 (Line 55-58): “… Beside participation in olaparib´s pharmacodynamic effect, BRCA mutations have a valuable clinical prognostic value, such as for guiding selection of patients eligible for secondary cytoreductive surgery within the treatment of recurrent ovarian cancer with liver metastases [25]”.

  1. Gallotta, V.; Conte, C.; D’Indinosante, M.; Capoluongo, E.; Minucci, A.; De Rose, A.M.; Ardito, F.; Giuliante, F.; Di Giorgio, A.; Zannoni, G.F.; et al. Prognostic factors value of germline and somatic brca in patients undergoing surgery for recurrent ovarian cancer with liver metastases. Eur. J. Surg. Oncol. J. Eur. Soc. Surg. Oncol. Br. Assoc. Surg. Oncol. 2019, 45, 2096–2102, doi:10.1016/j.ejso.2019.06.023.

Reviewer 3 Report

Authors demonstrated that olaparib, a potent poly(ADP-ribose) polymerase (PARP) inhibitor, potently inhibits the aldo-keto reductase AKR1C3 at clinically relevant concentrations. AKR1C3 is is the most active enzyme in the decrease of the intracellular concentration of pharmacologically active forms of anthracyclines and in anthracycline resistance. Therefore, olaparib is active alone and in combination with anthracyclines in cancer cells with AKR1C3 expression. These findings can be translated into therapies of cancers expressing AKR1C3.

The manuscript is well written. I have only some suggestions to improve the text. As the authors described, olaparib alone is used in cancer treatment in cancer cells harboring defects in DNA damage repair (BRCA mutations). Olaparib is studied also in the treatment of myelodysplastic syndrome and acute myeloid leukemia (AML) with defects in DNA repair (Faraoni I, Consaivo MI, Aloisio F, et al. Cancers 2019, 11, 1373; Maifrede S, Nieborowska-Skorska M, Sullivan-Teed K., et al. Blood 2018, 132(1), 67-77). Olaparib induces upregulation of death receptors-mediated apoptosis in AML cells (Meng XW, Koh BD, Zhang J-S, et al. J Biol Chem 2014, 289(30), 20543-20558; Faraoni I, Aloisio S, De Gabrieli A, et al. Cancer Lett 2018, 423, 127-138). Some abbreviations are not cleared in the first place of using (in Abstract aldo-keto reductase AKR1C3 should be used; on page 9 - empty vector (EV)- transfected HCT116 cells, it is cleared in Materials and Methods similarly as carcinoma cell lines used in abstract). Leukaemia, but leukemic and not leukaemic terms are used.

Author Response

We thank the Reviewer for the interest in our work and for helpful comments that will significantly improve the manuscript, and we have tried to do our best to respond to the point raised. The Reviewer has brought up some a good point, and we appreciate the opportunity to clarify our research. As indicated below, we have checked the comment provided by the Referee and have made necessary changes accordingly to the indications.

General comment:

C: Authors demonstrated that olaparib, a potent poly(ADP-ribose) polymerase (PARP) inhibitor, potently inhibits the aldo-keto reductase AKR1C3 at clinically relevant concentrations. AKR1C3 is is the most active enzyme in the decrease of the intracellular concentration of pharmacologically active forms of anthracyclines and in anthracycline resistance. Therefore, olaparib is active alone and in combination with anthracyclines in cancer cells with AKR1C3 expression. These findings can be translated into therapies of cancers expressing AKR1C3. The manuscript is well written. I have only some suggestions to improve the text.

Specific comments:

C1: As the authors described, olaparib alone is used in cancer treatment in cancer cells harboring defects in DNA damage repair (BRCA mutations). Olaparib is studied also in the treatment of myelodysplastic syndrome and acute myeloid leukemia (AML) with defects in DNA repair (Faraoni I, Consaivo MI, Aloisio F, et al. Cancers 2019, 11, 1373; Maifrede S, Nieborowska-Skorska M, Sullivan-Teed K., et al. Blood 2018, 132(1), 67-77). Olaparib induces upregulation of death receptors-mediated apoptosis in AML cells (Meng XW, Koh BD, Zhang J-S, et al. J Biol Chem 2014, 289(30), 20543-20558; Faraoni I, Aloisio S, De Gabrieli A, et al. Cancer Lett 2018, 423, 127-138).

A: Thank you for your comment. We have added the suggested information and respective references concerning the studies involving olaparib.

P.2 (Line 58-61): “Apart from its common indications, olaparib exhibits significant potential for the treatment of myelodysplastic syndrome and acute myeloid leukaemia (AML) with defects in DNA repair [26,27], inducing the death receptors-mediated apoptosis in AML cells [28,29].”

References:

  1. Faraoni, I.; Consalvo, M.I.; Aloisio, F.; Fabiani, E.; Giansanti, M.; Di Cristino, F.; Falconi, G.; Tentori, L.; Di Veroli, A.; Curzi, P.; et al. Cytotoxicity and Differentiating Effect of the Poly(ADP-Ribose) Polymerase Inhibitor Olaparib in Myelodysplastic Syndromes. Cancers (Basel). 2019, 11, 1373, doi:10.3390/cancers11091373.
  2. Maifrede, S.; Nieborowska-Skorska, M.; Sullivan-Reed, K.; Dasgupta, Y.; Podszywalow-Bartnicka, P.; Le, B.V.; Solecka, M.; Lian, Z.; Belyaeva, E.A.; Nersesyan, A.; et al. Tyrosine kinase inhibitor–induced defects in DNA repair sensitize FLT3(ITD)-positive leukemia cells to PARP1 inhibitors. Blood 2018, 132, 67–77, doi:10.1182/blood-2018-02-834895.
  3. Faraoni, I.; Aloisio, F.; De Gabrieli, A.; Consalvo, M.I.; Lavorgna, S.; Voso, M.T.; Lo-Coco, F.; Graziani, G. The poly(ADP-ribose) polymerase inhibitor olaparib induces up-regulation of death receptors in primary acute myeloid leukemia blasts by NF-κB activation. Cancer Lett. 2018, 423, 127–138, doi:https://doi.org/10.1016/j.canlet.2018.03.008.
  4. Meng, X.W.; Koh, B.D.; Zhang, J.-S.; Flatten, K.S.; Schneider, P.A.; Billadeau, D.D.; Hess, A.D.; Smith, B.D.; Karp, J.E.; Kaufmann, S.H. Poly(ADP-ribose) polymerase inhibitors sensitize cancer cells to death receptor-mediated apoptosis by enhancing death receptor expression. J. Biol. Chem. 2014, 289, 20543–20558, doi:10.1074/jbc.M114.549220.

C2: Some abbreviations are not cleared in the first place of using (in Abstract aldo-keto reductase AKR1C3 should be used; on page 9 - empty vector (EV)- transfected HCT116 cells, it is cleared in Materials and Methods similarly as carcinoma cell lines used in abstract). Leukaemia, but leukemic and not leukaemic terms are used.

A: Thank you for pointing this out. We have cleared all the abbreviations.

P.1 (Line 15-21): “Abstract… Incubation experiments with the most active recombinant CREs showed that olaparib is a potent inhibitor of the aldo-keto reductase 1C3 (AKR1C3) enzyme. Subsequent inhibitory assays in the AKR1C3-overexpressing cellular model transfected human colorectal carcinoma HCT116 cells demonstrated that olaparib significantly inhibits AKR1C3 at the intracellular level. Consequently, molecular docking studies supported these findings and identified the possible molecular background of the interaction. Drug combination experiments in HCT116, human liver carcinoma HepG2, and leukemic KG1α cell lines…”

Regarding EV abbreviation, it was mentioned first time in subsection 2.3 on page 4, where respective abbreviations were introduced:

Line 124: “assay in AKR1C3 and empty vector-transfected HCT116 cells (HCT116-AKR1C3 and HCT116-EV, respectively)”.

Also, we clarified it on page 9:

Line 225: “… differential response in AKR1C3- vs. empty vector-transfected HCT116 cells to the Daun combination with olaparib.”

For terms regarding leukaemia/leukemic, we are sorry, but we do not understand, as these terms are used in accordance with your commentary in original manuscript. In addition, original manuscript underwent professional language edition by Elsevier and should be fully matching requirement for language quality.

Reviewer 4 Report

Defining mechanisms of drug resistance in different cancer is crucial. While there are several causes of resistance, each pathway needs to be delineated. In order to demonstrate the role of AKR1C3 in drug resistance and the importance of targeting this enzyme in different cancers, more experiments have to be performed.
1- Enzyme activities of CREs in newly diagnosed patients and those who are resistant to anthracyclines need to be measured, and using one cell line for each disease cannot be a good model for expanding the results.
2-IC50 of Daunorubicin before and after overexpression of AKR1C3 needs to be measured in each cell line.
3- IC50 of Olaparib before and after overexpression of AKR1C3 needs to be measured in each cell line.
4- More cellular and molecular experiments have to be done in order to demonstrate the efficacy and mechanism of the combination therapy.

Author Response

We thank the Reviewer for the interest in our work and for helpful comments that will significantly improve the manuscript, and we have tried to do our best to respond to the points raised. The Reviewer has brought up some good points, and we appreciate the opportunity to clarify our research objectives and results. As indicated below, we have checked the general and specific comments provided by the Referee and have made necessary changes accordingly to the indications.

General comment:

C: Defining mechanisms of drug resistance in different cancer is crucial. While there are several causes of resistance, each pathway needs to be delineated. In order to demonstrate the role of AKR1C3 in drug resistance and the importance of targeting this enzyme in different cancers, more experiments have to be performed.

Specific comments:

C1: 1- Enzyme activities of CREs in newly diagnosed patients and those who are resistant to anthracyclines need to be measured and using one cell line for each disease cannot be a good model for expanding the results.

A: Thank you for your commentary. We are sorry, but within major revision, it is impossible to start cooperation with clinicians from several oncological fields, collect huge number of samples and measure the expression and activities of CREs. This is job for several years. However, to comply with your commentary as best as possible, we performed additional literature survey and found several pieces of information on increased expression of CREs in anthracycline resistant versus in sensitive patients. Following sentence was added to Introduction:

Line 47-49: Importantly, AKR1C3 together with other important CREs (AKR1B10, CBR1) are overexpressed in oncological patients resistant to ANTs, which confirms the critical role of reductive metabolism in ANT resistance [2,15–18]*.

*References

  1. Piska, K.; Koczurkiewicz, P.; Bucki, A.; Wójcik-PszczoÅ‚a, K.; KoÅ‚aczkowski, M.; PÄ™kala, E. Metabolic carbonyl reduction of anthracyclines - role in cardiotoxicity and cancer resistance. Reducing enzymes as putative targets for novel cardioprotective and chemosensitizing agents. Invest. New Drugs 2017, 35, 375–385, doi:10.1007/s10637-017-0443-2.
  2. Matsunaga, T.; Wada, Y.; Endo, S.; Soda, M.; El-Kabbani, O.; Hara, A. Aldo-Keto Reductase 1B10 and Its Role in Proliferation Capacity of Drug-Resistant Cancers. Front. Pharmacol. 2012, 3, 5, doi:10.3389/fphar.2012.00005.
  3. Bortolozzi, R.; Bresolin, S.; Rampazzo, E.; Paganin, M.; Maule, F.; Mariotto, E.; Boso, D.; Minuzzo, S.; Agnusdei, V.; Viola, G.; et al. AKR1C enzymes sustain therapy resistance in paediatric T-ALL. Br. J. Cancer 2018, 118, 985–994, doi:10.1038/s41416-018-0014-0.
  4. Varatharajan, S.; Abraham, A.; Zhang, W.; Shaji, R. V; Ahmed, R.; Abraham, A.; George, B.; Srivastava, A.; Chandy, M.; Mathews, V.; et al. Carbonyl reductase 1 expression influences daunorubicin metabolism in acute myeloid Eur. J. Clin. Pharmacol. 2012, 68, 1577–1586, doi:10.1007/s00228-012-1291-9.
  5. Marin, J.J.G.; Briz, O.; Rodríguez-Macias, G.; Díez-Martín, J.L.; Macias, R.I.R. Role of drug transport and metabolism in the chemoresistance of acute myeloid leukemia. Blood Rev. 2016, 30, 55–64, doi:10.1016/j.blre.2015.08.001.

C2:  IC50 of Daunorubicin before and after overexpression of AKR1C3 needs to be measured in each cell line.

A: Thank you for your commentary. The combination index method of Chou-Talalay is based on the mathematical algorithm, which considers IC50 values of both drugs alone for calculating combination outcome. This is valid for both non-constant and constant ratio experimental setup (Chou, 2006)*. IC50s of daunorubicin were calculated by CompuSyn software during the analysis of combination data. Daunorubicin IC50 in HCT116-AKR1C3 cells was 0.808 ± 0.049 µM, while IC50 in HCT116-EV cells was 0.494 ± 0.039 µM. These values reflect lower sensitivity of AKR1C3-overexpressing cells to daunorubicin, which is in accordance with recognized role of the enzyme in daunorubicin resistance. At the same time, these results are in a good correlation with our previously published data (Novotná et al., 2018a; Novotná et al., 2018b; Novotná et al., 2020)*. The IC50s come from data points presented in Fig 6a, 6b (light grey columns); data from this figure were first transformed into Fa-drug effect data and then inserted into CompuSyn software generating above mentioned values. As Fa-CI plots (Fig. 6c, 6d) are considered the most appropriate way of the presentation of drug combinations outcomes, IC50s are not necessary to be mentioned in the manuscript.

*References

Chou, T.-C. Theoretical basis, experimental design, and computerized simulation of synergism and  antagonism in drug combination studies. Pharmacol. Rev. 2006, 58, 621–681, doi:10.1124/pr.58.3.10.

Novotná, E.; Büküm, N.; Hofman, J.; Flaxová, M.; Kouklíková, E.; Louvarová, D.; Wsól, V. Roscovitine and purvalanol A effectively reverse anthracycline resistance mediated by the activity of aldo-keto reductase 1C3 (AKR1C3): A promising therapeutic target for cancer treatment. Biochem. Pharmacol. 2018a, 156, 22–31, doi:10.1016/j.bcp.2018.08.001.

Novotná, E.; Büküm, N.; Hofman, J.; Flaxová, M.; Kouklíková, E.; Louvarová, D.; Wsól, V. Aldo-keto reductase 1C3 (AKR1C3): a missing piece of the puzzle in the dinaciclib interaction profile. Arch. Toxicol. 2018b, 92, 2845–2857, doi:10.1007/s00204-018-2258-0.

Novotná, E.; Morell, A.; Büküm, N.; Hofman, J.; Danielisová, P.; Wsól, V. Interactions of antileukemic drugs with daunorubicin reductases: could reductases affect the clinical efficacy of daunorubicin chemoregimens? Arch. Toxicol. 2020, 94, 3059–3068, doi:10.1007/s00204-020-02818-y.

C3: IC50 of Olaparib before and after overexpression of AKR1C3 needs to be measured in each cell line.

A: Thank you for your commentary. As mentioned in reply to your commentary C2, CompuSyn software require dose-response data for both drugs alone to be able to calculate CI values of drug combination. As mentioned in 2.4, olaparib concentration used in drug combinations “exerted negligible toxicity in tested cell lines”. CompuSyn calculated olaparib’s IC50 values equal to 202 ± 46 and 186 ± 51 µM in HCT116-AKR1C3 and HCT116-EV cells, respectively. Again, data of Fa-CI plots originate also from these data and values, so there is no need to mention them separately in the manuscript. However, we realized two mistakes concerning discussed issue, which have been overlooked during preparation of manuscript: 1) Although in the methodological section 4.9 of original manuscript, it was mentioned that olaparib’s effect on viability was measured alone in cellular models, there were non-actualized incorrect concentration ranges for transfected HC116 sublines and HepG2 cells. 2) Information about analysis of olaparib alone was missing in legends to Fig. 6 and Fig. 7, thus possibly confusing the readers. Both mistakes were corrected in revised manuscript.

C4: More cellular and molecular experiments have to be done in order to demonstrate the efficacy and mechanism of the combination therapy.

A: Thank you for your commentary. There can exist several mechanisms driving the synergy between olaparib and anthracyclines. In our study, we are not able to cover deciphering of all these possible mechanisms. Instead, we aimed to clarify, whether olaparib-mediated interference with daunorubicin metabolism might be one of the participating factors affecting their combination outcome. For this purpose, we used various appropriate in vitro techniques and provided solid pieces of evidence supporting this hypothesis. We believe that present manuscript contains sufficient experimental bacgkround to provide initial insight into this interesting issue. In vivo studies will be needed to verify clinical importance of observed interaction and its impact on combination outcome, but they are planned to be a part of future separate study. Regarding efficacy of the combination, it has already been demonstrated in previous experimental studies (Park et al., 2018; Eetezadi et al., 2018)*. In addition, beneficial combination outcomes led to the introduction of this strategy into clinical evaluation (Del Conte et al., 2014)*.

*References

Park, H.J.; Bae, J.S.; Kim, K.M.; Moon, Y.J.; Park, S.H.; Ha, S.H.; Hussein, U.K.; Zhang, Z.; Park, H.S.; Park, B.H.; et al. The PARP inhibitor olaparib potentiates the effect of the DNA damaging agent doxorubicin in osteosarcoma. J. Exp. Clin. Cancer Res. 2018, 37, 107, doi:10.1186/s13046-018-0772-9.

Eetezadi, S.; Evans, J.C.; Shen, Y.T.; De Souza, R.; Piquette-Miller, M.; Allen, C. Ratio-Dependent Synergism of a Doxorubicin and Olaparib Combination in 2D and Spheroid Models of Ovarian Cancer. Mol. Pharm. 2018, 15, 472–485, doi:10.1021/acs.molpharmaceut.7b00843.

Del Conte, G.; Sessa, C.; Von Moos, R.; Viganò, L.; Digena, T.; Locatelli, A.; Gallerani, E.; Fasolo, A.; Tessari, A.; Cathomas, R.; et al. Phase i study of olaparib in combination with liposomal doxorubicin in patients with advanced solid tumours. Br. J. Cancer 2014, 111, 651–659, doi:10.1038/bjc.2014.345.
